# A High–Efficiency Side–Scan Sonar Simulator for High–Speed Seabed Mapping

**DOI:** 10.3390/s23063083

**Published:** 2023-03-13

**Authors:** Xiangjian Meng, Wen Xu, Binjian Shen, Xinxin Guo

**Affiliations:** 1Institute of Deep–Sea Science and Engineering, Chinese Academy of Sciences, Sanya 572000, China; mengxj@idsse.ac.cn (X.M.); wxu@zju.edu.cn (W.X.); shenbj@idsse.ac.cn (B.S.); 2University of Chinese Academy of Sciences, Beijing 100049, China; 3Ocean College, Zhejiang University, Zhoushan 316021, China

**Keywords:** side–scan sonar, sonar simulator, echo signal fitting algorithm, modeling simplification algorithm, new energy function

## Abstract

Side scan sonar (SSS) is a multi–purpose ocean sensing technology, but due to the complex engineering and variable underwater environment, its research process often faces many uncertain obstacles. A sonar simulator can provide reasonable research conditions for guiding development and fault diagnosis, by simulating the underwater acoustic propagation and sonar principle to restore the actual experimental scenarios. However, the current open–source sonar simulators gradually lag behind mainstream sonar technology; therefore, they cannot be of sufficient assistance, especially due to their low computational efficiency and unsuitable high–speed mapping simulation. This paper presents a sonar simulator based on a two–level network architecture, which has a flexible task scheduling system and extensible data interaction organization. The echo signal fitting algorithm proposes a polyline path model to accurately capture the propagation delay of the backscattered signal under high–speed motion deviation. The large–scale virtual seabed is the operational nemesis of the conventional sonar simulators; therefore, a modeling simplification algorithm based on a new energy function is developed to optimize the simulator efficiency. This paper arranges several seabed models to test the above simulation algorithms, and finally compares the actual experiment results to prove the application value of this sonar simulator.

## 1. Introduction

Side–scan sonar (SSS) is an underwater acoustic instrument widely used in several applications [1], such as ocean engineering assistance, shipwreck rescue, and military target detection. The acoustic sensor (also called a transducer) is the main component of the sonar system used to perform sound transmission and receival. In the design of such sensors, due to the complexity of the application environment, theoretical calculations cannot guarantee satisfactory performance, while field experiments are also difficult to carry out in the early stage of development [2]. For the purpose of design verification, in addition to the accumulation of practical experiences, researchers usually have to resort to certain analysis tools, e.g., a sonar simulator.

A sonar simulator is a numerical simulation technology that can provide a virtual experimental environment to support design, performance prediction, and fault diagnosis, which is especially useful for a new–principle sonar [3,4,5]. For instance, Bouxsein modeled the acoustic scattering characteristics of complex geometric surfaces, and then established an underwater simulation environment to explore the underwater automatic obstacle avoidance technology [6]. This low–cost approach with sufficient simulation results enabled him to summarize the sensor optimization theory of obstacle avoidance. Similarly, Sung proposed an underwater target search method using an acoustic imaging simulator and a convolutional neural network (CNN), which overcomes the difficulties of acquiring multi–view sonar images in unstable ocean environments [7].

The Synthetic Image Generator for Modeling Active Sonar (SIGMAS) developed by the NATO (North Atlantic Treaty Organization) Underwater Research Center (NURC) is the most advanced SSS simulator at present, which adopts both a finite element model (FEM) and a boundary element model (BEM) to accelerate the simulation performance [8]. The improved version of SIGMAS+, optimized by a graphics processing unit (GPU), constructs different render stages required for the final image and more refined effects, such as the sensor’s point spread function (PSF) and fast image correlation [9]. However, SIGMAS/SIGMAS+ is limited in accessibility. While some other sonar simulators are available for individual conventional systems, they typically lack a flexible data structure.

Research on sonar simulators often has to solve the mutual constraint between simulation fidelity and computational efficiency [10,11]. Essentially, echo signal fitting serves to calculate the acoustic characteristics of the scattering points in the scanning area, so the total point number is the main factor affecting the simulation performance. Such points, set only with spatial coordinates, are usually combined into a mesh structure using specific topologies, such as triangular meshes (TMs). Riordan adopted a geometric element deletion method to perform multi–resolution TMs, in which mesh elements not contributing to the final imaging are deleted before reaching the rendering pipeline, thus greatly saving computing resources while retaining valuable acoustic information [12].

A related modeling simplification algorithm has been developed following graphics rendering technology, where a TM not only records the spatial coordinates of the point cloud, but also topological information that is not meaningful for acoustic simulation [13,14,15,16]. Liu proposed a dual–mode scheme combining TMs and the point cloud, in which TMs are used as a base model for performing simplification processing, and the extracted point cloud later participates in echo estimation [17]. However, the methods only focus on geometric structure features, ignoring the influences arising from the underlying acoustic principle.

This paper presents a sonar simulator that can support the research requirements of advanced SSS. In order to provide convenience for the research community, it is open–source. The main contributions are summarized as follows:A two–level network architecture is developed, which can reasonably modularize the cumbersome simulation process to form a flexible operational network. It can support mainstream SSS engineering as well as module expansion for future research directions, thus being more advanced than the current open–source solutions.Different from the conventional sonar simulator using a stop&hop path model, an echo signal fitting algorithm based on the polyline path model is proposed, which can restore accurate propagation parameters of the backscattered signal to adapt the high–speed mapping simulation. Moreover, the Doppler effect is also accounted to achieve high–fidelity echo calculations.Avoiding the disadvantages of graphics rendering technology, more efficient point cloud is fully applied instead of redundant TMs. A modeling simplification algorithm based on a new energy function is proposed, which fully considers the acoustic principle and identifies the model structure sensitive to underwater acoustic signals, and then eliminates the low–value scattering points to accelerate the simulation performance on a large–scale virtual seabed.

The rest of this paper is organized as follows: Section 2 introduces the sonar simulator framework in terms of a two–level network architecture; Section 3 presents the echo signal fitting algorithm; Section 4 presents the modeling simplification algorithm based on a new energy function; Section 5 applies the lake experiment results to justify the performance of the developed sonar simulator; and Section 6 concludes this paper.

## 2. Sonar Simulator Framework Based on a Two–Level Network Architecture

This section starts by analyzing the operation principle of conventional SSS systems, and then clarifies the functional requirements to introduce the simulator framework.

### 2.1. Sonar Principle

The acoustic sensors are installed on both sides of the sonar platform, which radiates the acoustic beam to the seabed at a fixed frame rate and collects the echo signal to generate continuous strip–shaped images. These images reflect the topographic features and acoustic information in the scanning area. Then, the upper computer combines the navigation information to form a complete seabed mapping image [18,19].

In Figure 1, a signal–beam side–scan sonar forms a narrow receiving beam of $\theta_{-3\mathrm{dB}}$ (in degrees) along the azimuth, which is determined by the length L and operation frequency $f_{c}$ of the sensor as
(1)$$\theta_{-3\mathrm{dB}}=0.88\times \frac{c}{L\times f_{c}}\times \frac{180{}^{\circ}}{\pi }$$
where c is the sound speed in the water. $\theta_{-3\mathrm{dB}}$ is the angular beamwidth in elevation, which is often rather large to ensure strip coverage [20]. The receiving beam footprint has a width of $S_{\theta_{-3\mathrm{dB}}}$ at the maximum operation range $R_{sss}$ along the navigation direction.

Hence, in order to ensure the continuity of adjacent frames in strip maps, the navigation speed is constrained by
(2)$$v_{\max}\leq \frac{S_{\theta_{-3\mathrm{dB}}}\times c}{2\times R_{sss}}$$
which can only reach 2~6 knots [21].

With the increasing demand for oceanographic surveys, advanced SSS technologies have been developed to improve efficiency and/or accuracy. For example, multi–beam side–scan sonar based on dynamic aperture can maintain an ideal resolution for wide–swath imaging at high speed [22]; multi–pulse side–scan sonar can also achieve high–speed mapping by using continuous pulse modulation [23]; and multi–array synthetic aperture side–scan sonar can significantly improve the azimuth resolution [24]. All the above schemes need to be pre–validated by the sonar simulator with flexible engineering customization, such as sensor structure, mapping principle, and data scalability. However, most existing open–source sonar simulators can only adapt to single–beam side–scan sonar for low–speed mapping [25].

### 2.2. Two–Level Network Architecture

The simulator framework is shown in Figure 2. As a two–level network architecture, it allocates the whole operation process into a modular program at different stages, thus enabling flexible task scheduling.

The primary network is established according to the simulation function, and inside each module is a secondary network composed of different components that perform special processing segments and maintain dynamic operation relations.

Among them, the sonar engineering module is the entrance of the simulator, which is used to load the engineering parameters of the tested sonar system. Its secondary network includes a sensor structure, signal waveform, navigation tracking, and virtual seabed model. The mapping simulation module is the core part, which is used to calculate the dynamic physical interaction between the acoustic signal and the virtual seabed along the planned navigation track. Therefore, its secondary network, taking the ray–tracing theory [26,27,28] as the reference, establishes countless acoustic rays between the sensors and the scanning surface, as shown in Figure 3. Taking scattering point $S_{m}$ as an example, the total ray path can be used to estimate the phase property of its echo signal. The angle γ between the transmitting ray and $S_{m}$ can be used to estimate the target strength parameter. Similarly, the angle θ between the receiving ray and the receiving sensor can affect the response sensitivity, which is the basis for calculating the amplitude property.

Then, the acoustic imaging module collects these simulated echo signals to combine the strip–shaped images into a complete mapping image. Moreover, the simulated echo signal may provide a large number of controllable seabed imaging samples for underwater target recognition and image segmentation algorithm research [29].

## 3. Echo Signal Fitting Algorithm for High–Speed Mapping

Most sonar simulators adopt simplified simulation methods; e.g., the stop&hop path model, which prevents them from restoring echo accuracy [30]. This section proposes an echo signal fitting algorithm for high–speed mapping simulation in particular, including a polyline path model, echo signal computation, and Doppler effect correction.

### 3.1. Polyline Path Model

Figure 4 is the self–coordinate system established with the centroid of the sonar platform as the origin, in which the displacements along the x,y,z axis are sway, surge, and heave, respectively, while the rotation angles around the x,y,z axis are pitch β, roll η, and yaw α, respectively. Let us denote ${\left[\right]}^{T}$ as the transpose operation. The spatial position ${\left[x,y,z\right]}^{T}$ of the sensor can be calculated as
(3)$$\left[\begin{array}{c} x\\ y\\ z \end{array}\right]=M_{T}\left[\begin{array}{c} \Delta x\\ \Delta y\\ \Delta z \end{array}\right]+\left[\begin{array}{c} x_{A}\\ y_{A}\\ z_{A} \end{array}\right]$$
where ${\left[\Delta x,\Delta y,\Delta z\right]}^{T}$ is the relative position of the sensor in the self–coordinate system, and ${\left[x_{A},y_{A},z_{A}\right]}^{T}$ is the absolute position of the sonar platform in the geographic coordinate system. $M_{T}$ is the attitude–distance conversion matrix as
(4)$$\left[\begin{array}{ccc} \cos\alpha \cos\beta & -\sin\alpha \cos\beta & \sin\beta \\ \sin\alpha \cos\eta +\cos\alpha \sin\beta \sin\eta & \cos\alpha \cos\eta -\sin\alpha \sin\beta \sin\eta & -\cos\alpha \sin\eta \\ \sin\alpha \sin\eta -\cos\alpha \sin\beta \cos\eta & \cos\alpha \sin\eta +\sin\alpha \sin\beta \cos\eta & \cos\alpha \cos\eta \end{array}\right]$$

Similar to Figure 3, high–speed simulation mapping should be further described as a continuous motion process. Figure 5 shows that the platform sails from the position $P_{n-1}$ to the position $P_{n+1}$ along a polyline path, in which the transmitting sensor radiates the detection acoustic beam at the initial position $T_{n-1}$ and the receiving sensor continuously collects the echo signal along the track $\vec{R_{n-1}R_{n}}$. In the process of signal collection, each scattering point on the virtual seabed will determine its unique echo path by polling calculation. This model assumes that the attitude change only occurs at the polyline intersection $P_{n}$ and maintains this state until the next mapping stage $\vec{P_{n}P_{n+1}}$.

A scattering point $S_{m}$ exists in the scanning area, and $\vec{T_{n-1}S_{m}}+\vec{S_{m}R_{v}}$ is its echo path. Then, there must be a unique intermediate position $P_{v}$ where the echo propagation delay is equal to the navigation time as
(5)$$\frac{\left|\vec{R_{n-1}R_{v}}\right|}{v}=\frac{\left|\vec{T_{n-1}S_{m}}\right|+\left|\vec{S_{m}R_{v}}\right|}{c},\left|\vec{R_{n-1}R_{v}}\right|\leq v\cdot T$$

By constructing the spatial triangle $\Delta R_{n}R_{n-1}S_{m}$, there is a geometric relationship as
(6)$$\cos\left(∠R_{n}R_{n-1}S_{m}\right)=\frac{{\left|\vec{R_{n-1}R_{n}}\right|}^{2}+{\left|\vec{S_{m}R_{n-1}}\right|}^{2}-{\left|\vec{S_{m}R_{n}}\right|}^{2}}{2\times \left|\vec{R_{n-1}R_{n}}\right|\times \left|\vec{S_{m}R_{n-1}}\right|}$$

According to the triangle $\Delta R_{v}R_{n-1}S_{m}$, the acoustic ray $\vec{S_{m}R_{v}}$ can be solved as
(7)$${\left|\vec{S_{m}R_{v}}\right|}^{2}={\left|\vec{R_{n-1}R_{v}}\right|}^{2}+{\left|\vec{S_{m}R_{n-1}}\right|}^{2}-2\times \left|\vec{R_{n-1}R_{v}}\right|\times \left|\vec{S_{m}R_{n-1}}\right|\times \cos\left(∠R_{n}R_{n-1}S_{m}\right)$$
where relevant vectors can be determined through the known spatial coordinates, e.g.,
(8)$$\left|\vec{T_{n-1}S_{m}}\right|=\sqrt{{\left(x_{T_{n-1}}-x_{S_{m}}\right)}^{2}+{\left(y_{T_{n-1}}-y_{S_{m}}\right)}^{2}+{\left(z_{T_{n-1}}-z_{S_{m}}\right)}^{2}}$$

It is worth noting that the stop&hop path model simplifies this process: the sonar platform “stops” at position $P_{n-1}$ to instantly complete the transmission and collection of acoustic signals, and then suddenly “hops” to position $P_{n}$ to perform the next scanning action, which does not support restoring the real receiving beam footprint. Figure 6 establishes a virtual seabed and shows the difference between two such path models.

Moreover, the incident angle γ of the acoustic ray $\vec{S_{m}R_{v}}$ can be expressed as
(9)$$\gamma =\arccos\left(h_{S}/\left|\vec{S_{m}R_{v}}\right|\right)$$
where $h_{s}$ is the depth of the scattering point $S_{m}$. Similarly, direction of angle (DOA) [31] $\theta_{S}$ between the acoustic ray $\vec{S_{m}R_{v}}$ and the sensor $R_{v}$ can be expressed as
(10)$$\cos\left(\theta_{m,R_{v}}\right)=\left(\vec{S_{m}R_{v}}\cdot \vec{R_{n}R_{v}}\right)/\left(\left|\vec{S_{m}R_{v}}\right|\times \left|\vec{R_{n}R_{v}}\right|\right)$$

Figure 7 shows the simulation results of the above parameters, which will be used for computing the echo signal later.

### 3.2. Echo Signal Fitting

An echo signal can be regarded as the superposition of the response signals from scattering points, so the fitting process consists in estimating the amplitude and phase characteristics of those response signals [32].

#### 3.2.1. Amplitude Calculation

A sonar equation [33] can be used to estimate echo signal amplitude, e.g., from the echo level EL given by
(11)$$EL=SL-TL_{T}-TL_{R}+TS$$
where the propagation loss $TL_{T}/TL_{R}$ of the transmission path and the receiving path is determined by the acoustic diffusion mode and the echo propagation distance r as
(12)$$TL=20\log_{10}r+\alpha \times r$$
where the absorption loss factor α is a function of operation frequency. A seabed is a special reverberation scattering target and its target strength TS can be calculated as
(13)$$TS=10\mathrm{lg}\mu +10\mathrm{lg}\cos^{2}\gamma$$
where μ is the seabed geological scattering coefficient, and γ is the incident angle of each scattering point. Therefore, the amplitude of the response signal of a single scattering point $S_{m}$ can be obtained by
(14)$$A=10^{\left(EL+Mx\right)/20}$$
where $M_{x}$ is the response sensitivity of the receiving sensor.

#### 3.2.2. Array Signal Model

As shown in Figure 8, the receiving sensor is a uniform linear array (ULA) composed of 2E−1 piezoelectric ceramic elements with the same frequency response characteristics. In the Fresnel zone, the echo signal $x_{m}\left(t\right)$ of the scattering point $S_{m}$ on the receiving sensor can be expressed as
(15)$$x_{m}\left(t\right)=\sum_{e=1}^{2E-1}A_{m,e}\times s\left(t-u_{m,T}-u_{m,e}\right)$$
where s(t) is the transmitted signal. $u_{m,T}$ and $u_{m,e}$ are the transmission propagation delay and receiving propagation delay, respectively,
(16)$$u_{m,e}=\sqrt{r^{2}{}_{m,e}+{\left(\left(e-E\right)\cdot d\right)}^{2}-2\cdot r_{m,e}\cdot \left(e-E\right)\cdot d\cdot \cos\left(90{}^{\circ}-\theta_{m,e}\right)}/c$$

Summing up contributions from M scattering points, the echo signals y(t) on the receiving sensor can be expressed as
(17)$$y\left(t\right)=\sum_{m=1}^{M}\sum_{e=1}^{2E-1}A_{m,e}\times s\left(t-u_{m,T}-u_{m,e}\right)$$

Figure 9 is the algorithm architecture of the echo signal fitting. The wideband LFM signal can be decomposed into multiple narrowband components via fast Fourier transform (FFT) [34]. For each narrowband component, the time delay processing can be converted to phase shift compensation, i.e.,
(18)$$Y\left(f\right)=\sum_{m=1}^{M}\sum_{e=1}^{2E-1}\sum_{f}A_{m,e}\times e^{-j\cdot 2\pi \cdot f\cdot \left(u_{m,T}+u_{m,e}\right)}\times S\left(f\right),f=f_{0},f_{0}+\Delta f,\cdots ,f_{N}-\Delta f,f_{N}$$
where a narrow frequency interval Δf can improve waveform fidelity.

The doppler effect is also an inevitable problem in a high–speed acoustic mapping system, which refers to the signal frequency offset caused by the relative motion between the signal source and the receiver [35,36]. For the polyline path model in Figure 5, the frequency $f_{m,T}$ of the incident signal at the scattering point $S_{m}$ is
(19)$$f_{m,T}=\left(c\times f\right)/\left(c-v\times \cos\theta_{m,T}\right)$$
where $v\times \cos\theta_{m,T}$ is the velocity component along the transmitting ray $\vec{T_{n}S_{m}}$ in a single scanning period T. Similarly, the frequency $f_{m,e}$ of the response signal $x_{m}\left(t\right)$ from the scanning point $S_{m}$ is
(20)$$f_{m,e}=f_{S}+\left(v\times f_{m,T}\times \cos\theta_{m,e}\right)/c$$

In summary, the frequency domain expression of the sensor array signal is
(21)$$Y\left(f\right)=\sum_{m=1}^{M}\sum_{e=1}^{2E-1}\sum_{f}A_{m,e}\times e^{-j\cdot 2\pi \cdot \left[f_{m,T}\cdot u_{m,T}+f_{m,e}\cdot u_{m,e}\right]}\times S\left(f\right),f=f_{0},f_{0}+\Delta f,\cdots ,f_{N}-\Delta f,f_{N}$$
where the above result is equivalent to the output signals of the receiving sensor.

### 3.3. Simulation Experiment

This part adopts the virtual seabed in Figure 6a to conduct acoustic mapping simulation experiments on different side–scan sonar technologies to verify the effectiveness of the developed sonar simulator. For single–beam side–scan sonar, as shown in Figure 10, the low–speed mapping image shows that each curve target can be clearly distinguished, but its azimuth resolution degrades as the detection distance increases; the high–speed mapping image shows an apparent imaging fracture structure. The above simulation results are consistent with the actual characteristics of single–beam side–scan sonar.

Beamforming technology is the main principle of multi–beam side–scan sonar to achieve high–speed mapping. Figure 11 shows high–speed simulation mapping images under two different beamforming methods. The conventional beamforming technology based on multiple subarray sensors can indeed avoid imaging fracture structures at high speed, but its imaging false alarm may occur in the near field; the near–field dynamic focused beamforming technology, which is widely studied currently, can effectively avoid this phenomenon. In summary, the developed sonar simulator can faithfully feedback the characteristics of different side–scan sonar technologies; therefore, it can provide an effective virtual experimental condition for future sonar engineering.

## 4. Modeling Simplification Based on a New Energy Function

The echo signal fitting process of a large–scale virtual seabed is extremely computationally expensive. This section analyzes the geometric element deletion method based on the conventional energy function [37], and then proposes a new energy function exploiting the acoustic principle, which constructs a multi–resolution point cloud to simplify the virtual seabed, thus accelerating the simulation efficiency.

### 4.1. Conventional Energy Function

Hoppe et al. describe the virtual model as a piecewise linear mesh, consisting of triangular faces pasted together along their edges. Formally, a mesh can be defined as a pair of sets (K,V),where K is a simplicial complex representing the geometric elements, such as the vertices, edges, and faces, thus determining the topological type of the mesh; $V=\left\{v_{1},\ldots ,v_{m}\right\},v_{i}\in R^{3}$ is a set of vertex positions defining the shape of the mesh in $R^{3}$ (its geometric realization).
(22)$$E\left(K,V\right)=E_{dist}\left(K,V\right)+E_{rep}\left(K\right)+E_{spring}\left(K,V\right)$$

The above is the conventional energy function that evaluates the operation cost of the geometric element deletion method, and then simplifies elements with low fidelity to form a new multi–resolution topology $K\Rightarrow K^{\prime }$. The distance energy $E_{dist}$ is equal to the sum of squared distances from the points $X=\left\{x_{1},\ldots ,x_{n}\right\}$ to the geometric realization M=ϕv(|K|) as
(23)$$E_{dist}\left(K,V\right)=\sum_{i=1}^{n}d^{2}\left(x_{i},\phi v\left(\left|K\right|\right)\right)$$

Each of these distances is itself the solution to the minimization problem as
(24)$$d^{2}\left(x_{i},\phi v\left(\left|K\right|\right)\right)=\underset{b_{i}\in \left|K\right|}{\min}{\left\|x_{i}-\phi v\left(b_{i}\right)\right\|}^{2}$$
in which a naive approach to computing $b_{i}\in \left|K\right|\subset R^{m}$ is to project $x_{i}$ onto all of the faces of M, and then find the projection with minimal distance. The representation energy $E_{rep}$ eliminates meshes with a large number of vertices. It is set to be proportional to the number of vertices m of K and a user–selectable penalty weight $c_{rep}$.

The spring energy $E_{spring}$ places on each edge of the mesh a spring of rest length zero and spring constant κ as
(25)$$E_{spring}\left(K,V\right)={\sum_{\left\{j,k\right\}\in K}\kappa \left\|v_{j}-v_{k}\right\|}^{2}$$
which helps guide the optimization to a desirable minimum simplification cost.

Objectively, the conventional energy function focuses on the graphic rendering application of the solid geometric model, which pursues the global unified smoothness of simplified processing, but lacks the consideration of underwater acoustic principles.

Figure 12 shows the simulation results under different simplification scales. Although the simplified TM can always maintain its original appearance, it is the visual effect of triangular faces pasting. However, such triangular faces are extremely inefficient in the echo signal fitting process and the point cloud extracted from the simplified TM is impossible to restore to the original model appearance. Therefore, the conventional energy function cannot simplify the virtual seabed in an underwater acoustic simulation.

### 4.2. A New Energy Function Focusing on Acoustics

A new energy function is proposed as expression 28, which uses point cloud instead of TMs and fully considers the effect of underwater acoustic principles on modeling simplification. Due to the fact that the point cloud only needs to define a set of scattering point coordinates $V=\left\{v_{1},\ldots ,v_{m}\right\},v_{i}\in R^{3}$ of the virtual seabed, this avoids the redundant computing burden caused by topology operations.
(26)$$E\left(V\right)=w_{shadow}\times \left[w_{curvature}\cdot E_{curvature}\left(V\right)+w_{scattering}\cdot E_{scattering}\left(V\right)+w_{view}\cdot E_{view}\left(V\right)\right]$$

The occluded scattering points are unreachable to the acoustic ray, so this part can be directly simplified without causing echo signal distortion. As shown in Figure 13, along the same acoustic ray direction, the shadow area cannot meet the rule that the incidence angle $\gamma_{m}$ increases with the propagation distance. Formula (29) simplifies them by the shadow coefficient $w_{shadow\left(m\right)}$, which can eliminate the low–value points directly.
(27)$$w_{shadow\left(m\right)}=\left\{ \begin{array}{l} 0,\gamma_{m}<\max(\gamma_{1},\gamma_{m-1})\\ 1,\gamma_{m}>\max(\gamma_{1},\gamma_{m-1}) \end{array}\right.$$

The surface structure with local protrusions or depressions is sensitive to acoustic signals, which should especially be reserved. The curvature energy $E_{curvature}\left(V\right)$ is obtained from the Gaussian curvature G(V), which is calculated from the coordinate position of each scattering point as
(28)$$\left\{ \begin{array}{l} E_{curvature}\left(v_{i}\right)=10\log_{10}\frac{G\left(v_{i}\right)}{\max(G\left(V\right))},v_{i}=\left(x_{i},y_{i},z_{i}\right)\in V\subset R^{3}\\ G\left(v_{i}\right)=\frac{TW-U^{2}}{1+t^{2}+u^{2}},t=\frac{\partial z_{i}}{\partial x_{i}},u=\frac{\partial z_{i}}{\partial y_{i}},T=\frac{\partial^{2}z_{i}}{\partial x_{i}{}^{2}},U=\frac{\partial^{2}z_{i}}{\partial x_{i}\partial y_{i}},W=\frac{\partial^{2}z_{i}}{\partial y_{i}{}^{2}} \end{array}\right.$$

According to the Lambertian law [38], a large incident angle $\gamma_{m}$ of an acoustic ray tends to make large TS, and then the scattering energy $E_{scattering}\left(V\right)$ also follows the rule as
(29)$$E_{scattering}\left(v_{i}\right)=10\log_{10}\frac{\gamma_{m}\left(v_{i}\right)}{\max(\gamma_{m}\left(V\right))},v_{i}\in V\subset R^{3}$$

Generally, the receiving sensor is most sensitive to the scattering points near its acoustic axis. Thus, the axis energy $E_{axis}\left(V\right)$ is measured by the projection of the receiving beam in the scanning area. In practical applications, this new function needs to calculate the above energy components according to the real–time position of the sonar platform and the installation angle of the receiving sensor. For example, Figure 14 shows the simulation results under the same virtual model as Figure 12a.

In addition, this form also sets weight coefficients $w_{curvature},w_{scattering},w_{axis}$ to balance each energy component. Figure 15 shows the simulation results for comparison with Figure 12c–h. It is worth noting that, under the same simplified scale, the scattering points near the curve’s contour and the acoustic axis can always be effectively retained, which is also the main contribution of the echo signal fitting. Different from the global uniform simplification characteristics of the conventional energy function, this new function shows the identification characteristics of the underwater acoustic signal’s sensitive structure.

### 4.3. Simulation Experiment

Figure 16 shows the simulation imaging results of single–beam side–scan sonar at three knots, where the virtual seabed is simplified by two methods presented in Section 4.1 and Section 4.2. It can be concluded after comparison that, with the increase in the simplified scale, the seabed model processed by the traditional energy function gradually exhibits imaging distortion, while the new energy function always keeps the imaging results consistent with the original model. This is due to the fact that the new energy function can identify the local structures that are valuable for echo signal fitting, while the conventional energy function tends to be simplified to global homogenization.

## 5. Comparative Analysis of Simulation and Actual Experiments

This section compares the simulation mapping results with the actual experiment to verify the fidelity of the developed sonar simulator. The mapping target is the pier of the cross–lake bridge and the sonar simulator loads the same engineering parameters of the tested single–beam side–scan sonar using LFM signal as center frequency 400 kHz, bandwidth 30 kHz, and pulse width 5 ms.

As shown in Figure 17, the underwater structure of the bridge pier is four cylindrical supporting structures, and its mapping results can clearly identify the tail shadow left by the cylinder at the bottom of the lake, which is consistent with the actual imaging results.

Figure 18 shows the simulated imaging results under the artificially modulated motion deviation curve, which can comprehensively verify the response ability of the sonar simulator to the virtual experimental conditions. For example, under the effect of a sway deviation curve, piers A and B have imaging positioning errors greater than 10 m along the transverse distance; under the effect of the deviation curve, the image of region B is more evident than that of region A in the lake bottom slash. The above phenomena meet the preset experimental parameters.

## 6. Conclusions

This paper presents a sonar simulator with a two–level network architecture, which can provide engineering guidance and conduct virtual experiments. Different from most open–source sonar simulators, the developed simulator first proposes an original polyline path model to achieve accurate echo signal fitting in high–speed mapping simulations, which is also applicable to other underwater acoustic simulation methods. Meanwhile, it adopts a modeling simplification algorithm based on a new energy function, which is improved from the conventional energy function of computer graphics. The improvement is that the underwater acoustic theory is used as the operation rule of modeling simplification for the first time, and then the model structure that has evident value to the echo signal is protected in the processing process, so as to accelerate the operation efficiency without causing imaging distortion. Finally, compared with the mapping results in an actual lake experiment, simulations for a similar scenario model yield consistent features. Overall, it is shown that the developed simulator is capable of supporting relevant virtual experiments and engineering research.

It is worth noting that the simulator has a modular network structure, which can be readily adjusted to fit other sonar systems. In particular, the two–level architecture facilitates the addition of new function modules. For example, currently, it is assumed that the sound velocity is constant; a ray–tracing submodule [39] can be added to the mapping simulation module to handle a depth–dependent sound velocity profile.

## Figures and Tables

**Figure 1 sensors-23-03083-f001:**
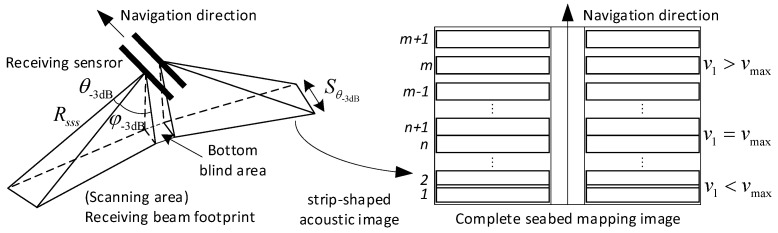
Imaging principle of the single–beam side–scan sonar.

**Figure 2 sensors-23-03083-f002:**
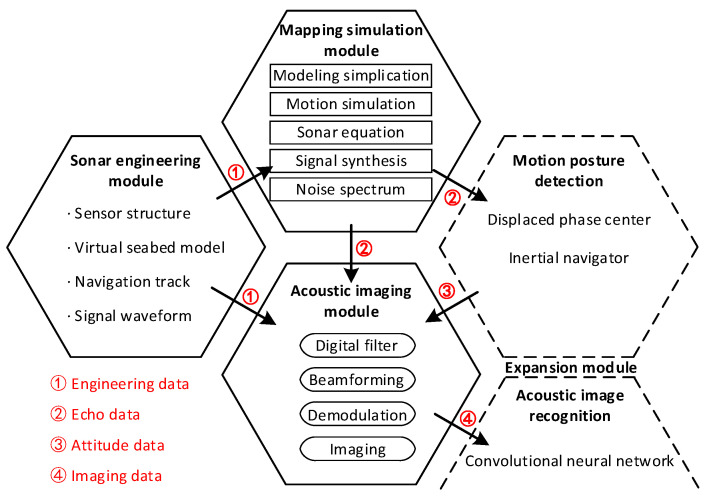
Sonar simulator framework as a two–level network architecture.

**Figure 3 sensors-23-03083-f003:**
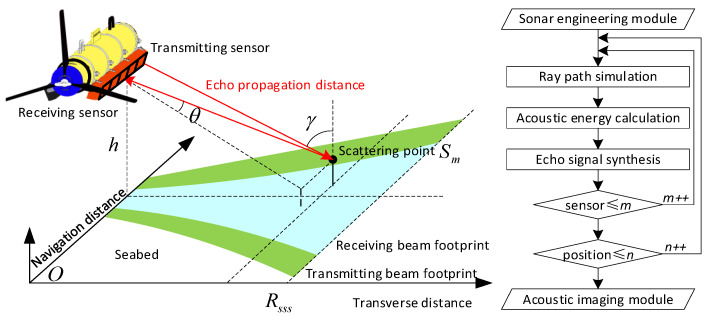
Schematic diagram of the ray–tracing algorithm.

**Figure 4 sensors-23-03083-f004:**
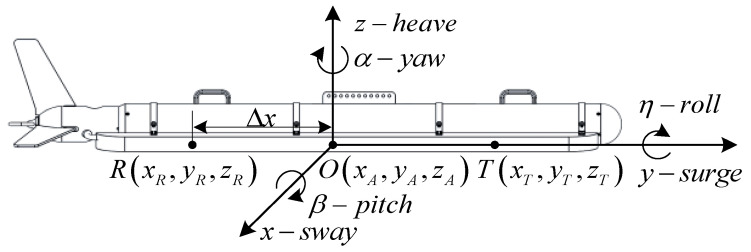
Self–coordinate system of the sonar platform.

**Figure 5 sensors-23-03083-f005:**
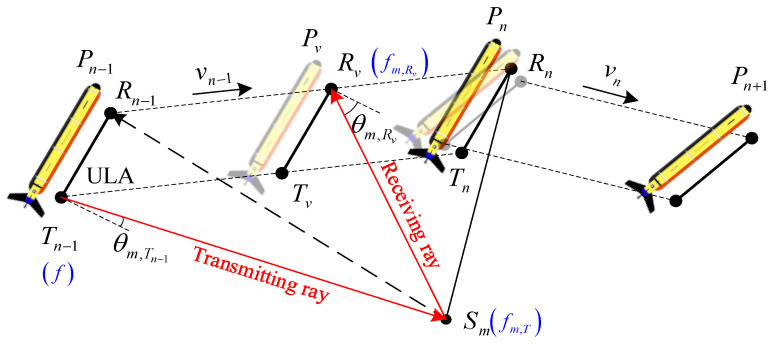
Scanning process in the polyline path model.

**Figure 6 sensors-23-03083-f006:**
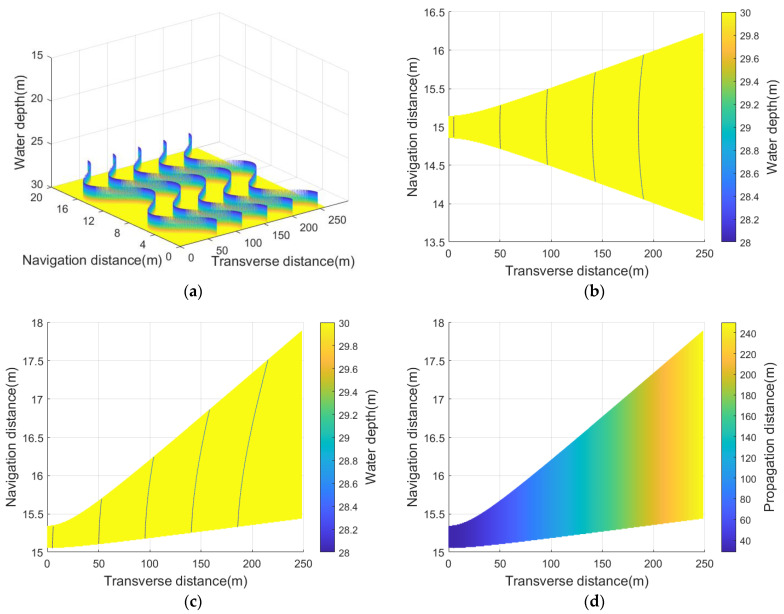
(**a**) A virtual seabed with 5 parallel harmonic curve targets; (**b**) the inaccurate receiving beam footprint of stop&hop path model; (**c**) the accurate receiving beam footprint of polyline path model; (**d**) acoustic propagation distance in the receiving beam footprint.

**Figure 7 sensors-23-03083-f007:**
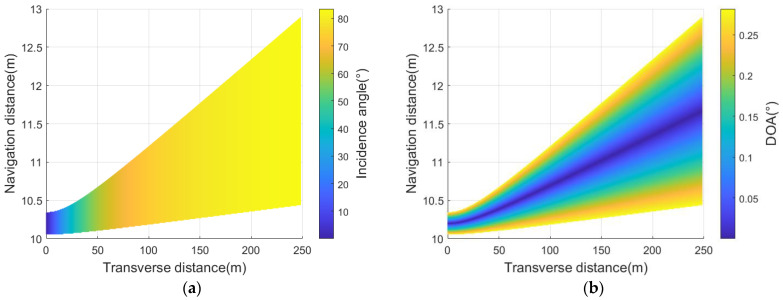
Echo parameters in the receiving beam footprint (Figure 6c): (**a**) incident angle; (**b**) DOA.

**Figure 8 sensors-23-03083-f008:**
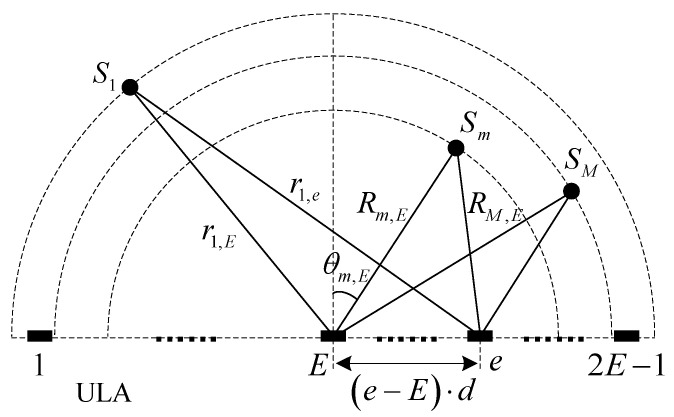
Sensor array and scattering point configuration.

**Figure 9 sensors-23-03083-f009:**
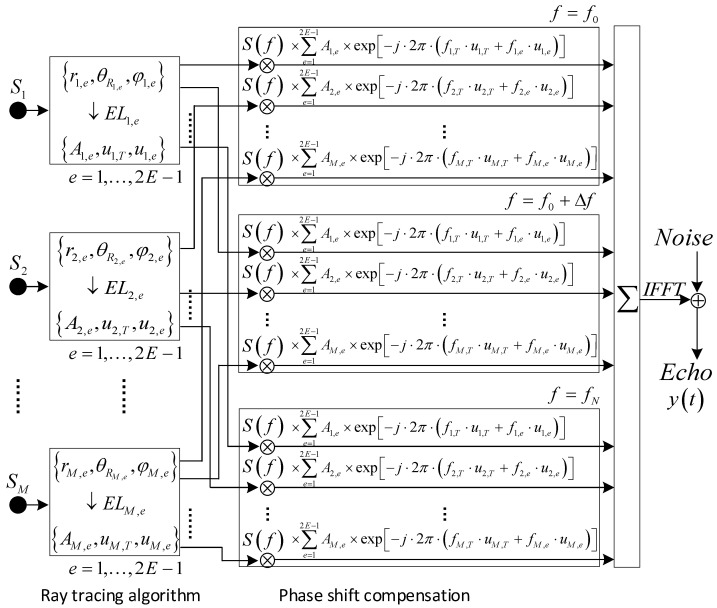
Echo signal fitting algorithm.

**Figure 10 sensors-23-03083-f010:**
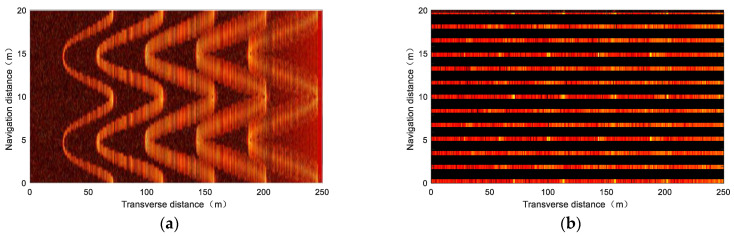
Simulation mapping images of the single–beam side–scan sonar used LFM signal as center frequency 400 kHz, bandwidth 30 kHz, and pulse width 5 ms: (**a**) at 3 knots; (**b**) at 10 knots.

**Figure 11 sensors-23-03083-f011:**
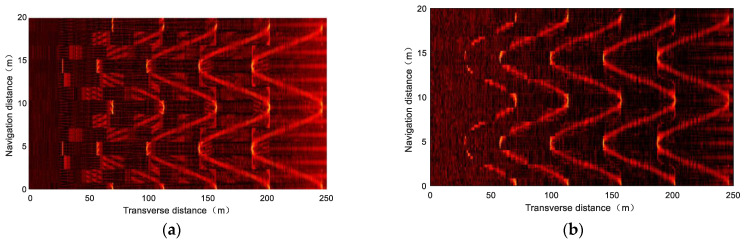
Simulation mapping images at 10knots: (**a**) conventional beamforming; (**b**) dynamic focused beamforming. The receiving sensor consists of 5 subarrays using LFM signal as center frequency 400 kHz, bandwidth 30 kHz, and pulse width 5 ms.

**Figure 12 sensors-23-03083-f012:**
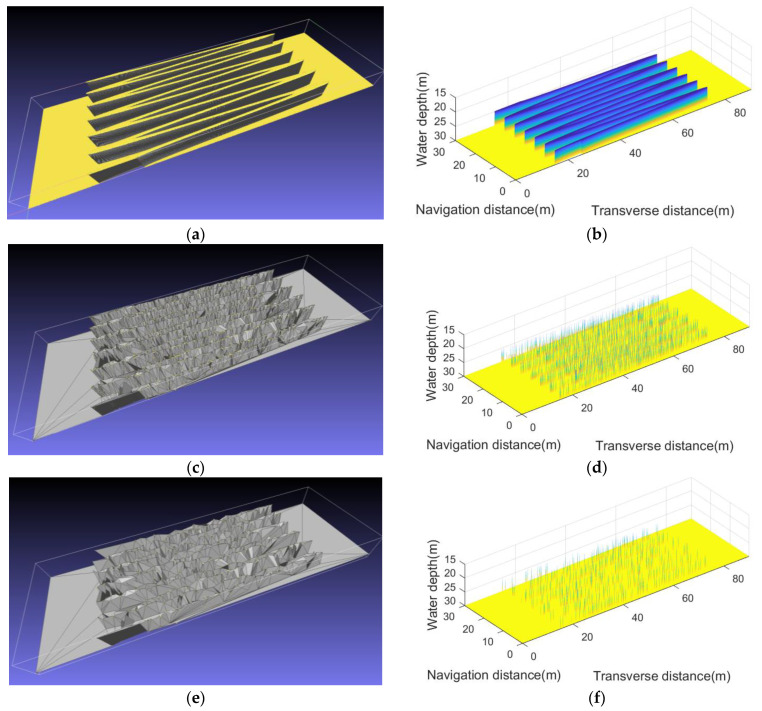
Simulation results under different simplification scales, where (**a**,**c**,**e**,**g**) are TM and (**b**,**d**,**f**,**h**) are the corresponding extracted point cloud: (**a**,**b**) are the original model composed of 42,936 data points; (**c**,**d**) are simplified to 3000 data points; (**e**,**f**) are simplified to 1000 data points; (**g**,**h**) are simplified to 176 data points.

**Figure 13 sensors-23-03083-f013:**
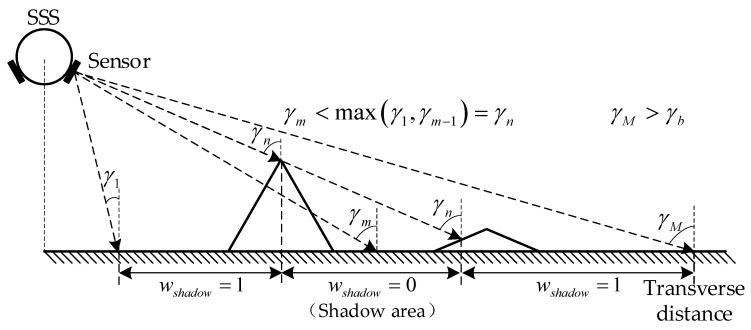
Geometric model for calculating shadow coefficient.

**Figure 14 sensors-23-03083-f014:**
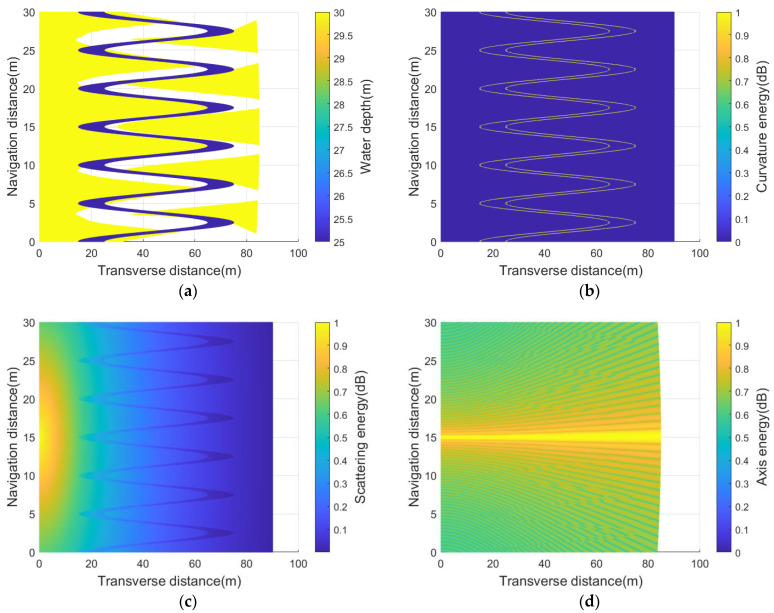
The receiving sensor ($\theta_{-3\mathrm{dB}}=0.56{}^{\circ},f_{c}=400\mathrm{kHz}$) looks down 30° horizontally at (0,10,30) coordinate: (**a**) the virtual seabed simplified by shadow coefficient $w_{shadow\left(m\right)}$; (**b**) the curvature energy $E_{curvature}\left(V\right)$; (**c**) the scattering energy $E_{scattering}\left(V\right)$; (**d**) the acoustic axis energy $E_{axis}\left(V\right)$.

**Figure 15 sensors-23-03083-f015:**
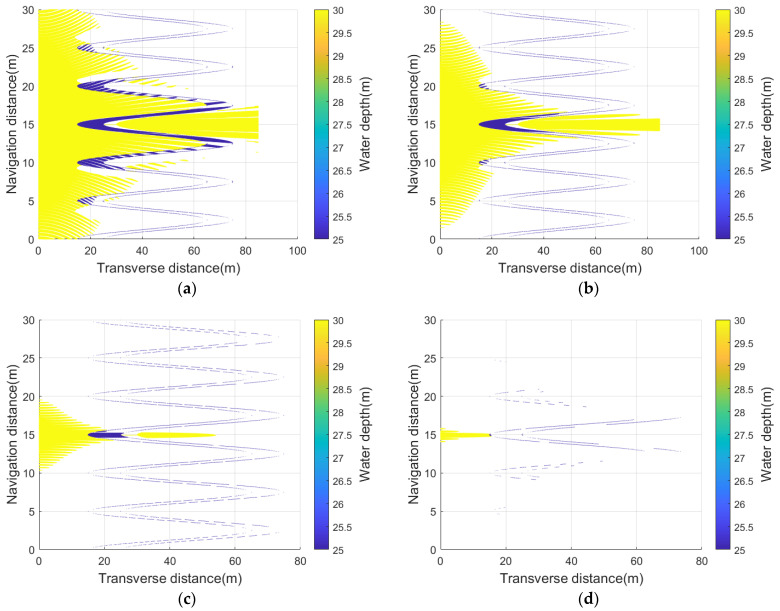
Scanning seabed samples processed under different simplification scales using a receiving sensor as center frequency 400 kHz, horizontal beam width 0.56°: (**a**) is simplified to 5125 data points; (**b**) is simplified to 3000 data points; (**c**) is simplified to 1000 data points; (**d**) is simplified to 176 data points.

**Figure 16 sensors-23-03083-f016:**
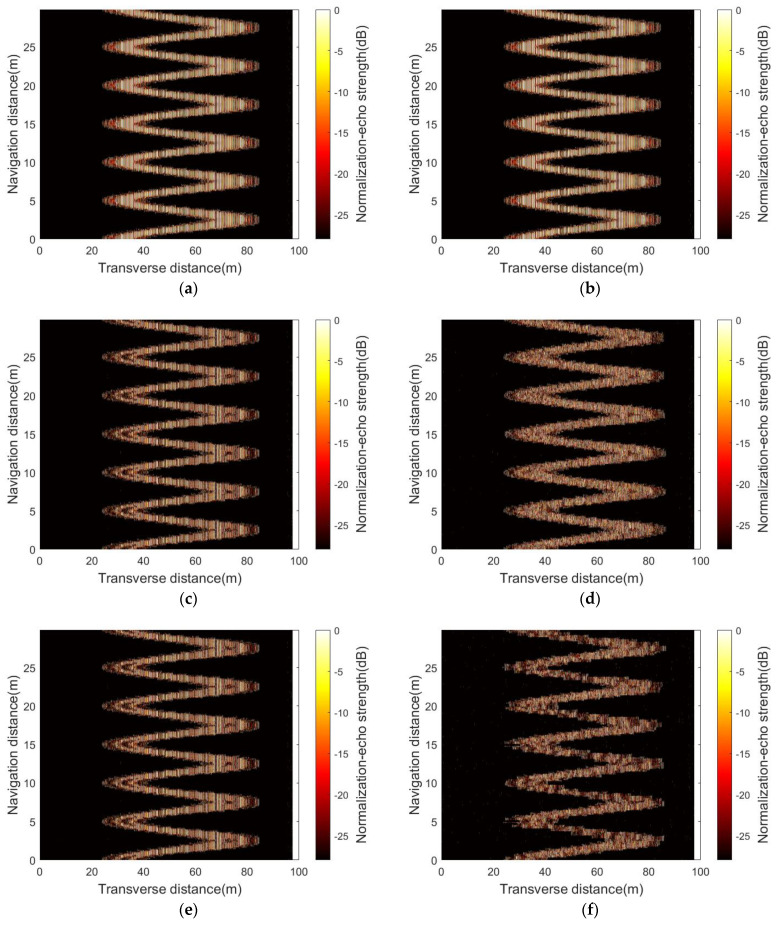
Simulation imaging results using single–beam side–scan sonar using LFM signal as center frequency 400 kHz, bandwidth 30 kHz, and pulse width 5 ms, where (**c**,**e**,**g**) are simplified by the new energy function and (**d**,**f**,**h**) are simplified by the conventional energy function: (**a**) is the simulation image of the original model; (**b**) is simplified to 5125 data points using the new energy function; (**c**,**d**) are simplified to 3000 data points; (**e**,**f**) are simplified to 1000 data points; (**g**,**h**) are simplified to 176 data points.

**Figure 17 sensors-23-03083-f017:**
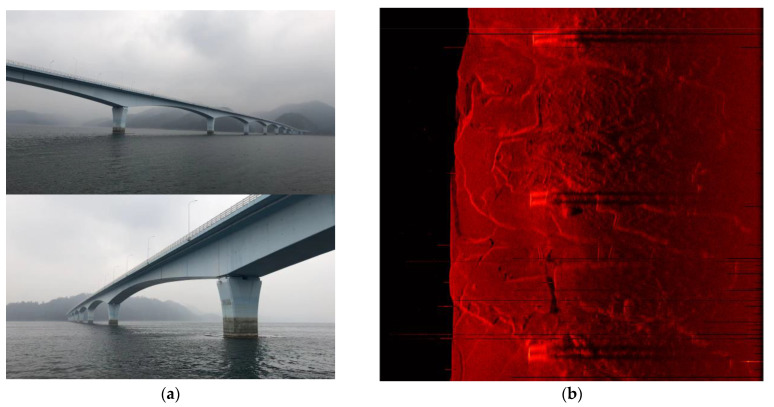
(**a**) Qiandao lake experimental environment in Hangzhou; (**b**) actual mapping results of bridge piers carried out in December 2022; (**c**) virtual experimental environment consisting of bridge piers and lake bottom slash; (**d**) simulation imaging result using the developed sonar simulator.

**Figure 18 sensors-23-03083-f018:**
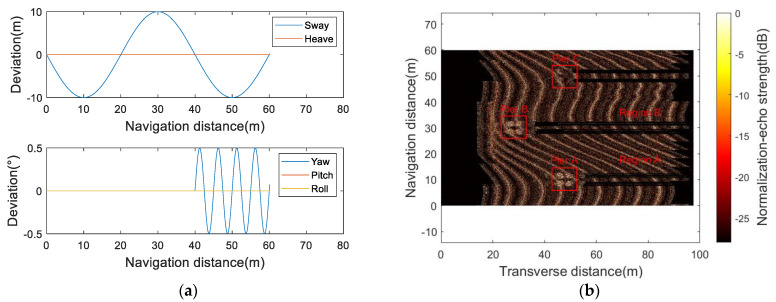
(**a**) Artificially modulated motion deviation curve; (**b**) simulation imaging result.

## Data Availability

The source code of the sonar simulator proposed in this paper is completely open–source and can be found at https://github.com/MENGXiang–jian/Side–scan–sonar–simulator.git (accessed on 9 September 2022). Any questions are welcome as feedback to the author.

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
