# Peer review of "A High–Efficiency Side–Scan Sonar Simulator for High–Speed Seabed Mapping"

_sensors, 2023, doi:10.3390/s23063083_

Round 1
Reviewer 1 Report
I reviewed the manuscript, sensors-2228009, “A High-Efficiency Side Scan Sonar Simulator for High-Speed Seabed Mapping”. Authors describe a new side scan sonar simulator for testing algorithms, etc. during sonar development. Authors claim that this simulator has acoustic fidelity and not limited to sonar equation so that the effects of propagation and reflection/scattering on the received signal are also modeled. The details of the analytical modeling is given in the manuscript. Authors also compare the simulation predictions with measurement data.
Comments:
1. Manuscript covers the presented work comprehensively. The manuscript is like a well written project report. Reviewer believes that the reported work is valuable, however the original contributions in the work are not clearly indicated. The quality of the manuscript will improve if the original contributions are clarified.
2. The manuscript is obviously proofread and the language is adequate. However, reviewer believes that the manuscript will benefit from another proof reading by a native english speaker.
Reviewer 2 Report
1. The authors present a sonar simulator based on a two-level network architecture, which has flexible task scheduling system and extensible data interaction organization. Echo signal fitting algorithm proposes a polyline path model to accurately capture the propagation delay of the backscattered signal under high-speed motion deviation. The large-scale virtual seabed is the operational nemesis of the conven-tional sonar simulators, then a modeling simplification algorithm based on a new energy function is developed to optimize the simulator efficiency
2. In the figure 3, schematic diagram of ray tracing algorithm should be demonstrated in detail.
3. In the figure 5, scanning process in the polyline path model should be demonstrated in detail.
4. The manuscript has 31 equations; the number of the equations should be decreased.
5. Revise the English thoroughly before submission.
Reviewer 3 Report
The paper presents SideScan Sonar simulator.
The main advantages are:
1) two-level network architecture;
2) echo signal fitting algorithm;
3) simplified modeling based on the modified energy function.
Significant graphical material is given to compare the quality of the computation grid.
This work will improve:
1) numerical comparison of the simulator with similar simulators results
For example, in the form of a table
The classic simulator, SIGMA+, and our simulator
Comparison parameters: computational speed, computational resources, quality of the restored bottom structure, and so on
2) similarly, comparison of echo signal fitting algorithms
Here you can do the same.
The work is certainly relevant and should be published after revision.
